# Motion Planning in UAV-Aided Data Collection with Dynamic Jamming

**Binbin Wu, Bangning Zhang, Wenfeng Ma \*, Chen Xie, Daoxing Guo and Hao Jiang** 

College of Communications Engineering, Army Engineering University of PLA, Nanjing 210007, China
\* Correspondence: 13913945193@139.com

**Abstract:** Unmanned-aerial-vehicle (UAV)-aided data collection for Internet of Things applications has attracted increasing attention. This paper investigates motion planning for UAV collecting low-power ground sensor node (SN) data in a dynamic jamming environment. We targeted minimizing the flight energy consumption via optimization of the UAV trajectory while considering the indispensable constraints which cover the collection data demodulation threshold, obstacle avoidance, data collection volume, and motion principle. Firstly, we formulate the UAV-aided data collection problem as an energy consumption minimization problem. To solve this nonconvex optimization problem, we rewrite the original problem by introducing relaxation variables and constructing equivalence constraints to obtain a new relaxation convex problem, which can be solved iteratively using the successive convex approximation (SCA) method. However, SCA is susceptible to initial values, especially in dynamic environments where fixed initial values may lead to a wide range of results, making it difficult to obtain a truly optimal solution to the optimization problem. To solve the initial value problem in dynamic environments, we further propose a communication-flight-corridor(CFC)-based initial path generation method to improve the reliability and convergence speed of the SCA method by constructing reliable communication regions and resilient secure paths in real time. Finally, simulation results validate the performance of the proposed algorithm compared to the benchmark algorithms under different parameter configurations.

**Keywords:** UAV; motion planning; data collection; dynamic jammings; SCA; CFC



## 1. Introduction

### 1.1. Background

Recently, communication supported by unmanned aerial vehicles (UAVs) has become an attractive technology in the field of wireless sensor networks (WSNs) for highly efficient data collection [1–3]. Most sensor nodes (SNs) are battery powered and deployed in a specific geographical area to sense and transmit the sensed information. At times, it is impractical to let these devices transmit or relay their sensed data to the base station through a multi-hop relay because of the high consumption of transmission energy. In the worst-case scenario, they may be out of range of each other's transmissions, particularly in the case of interference. As a result, it can be very challenging to gather sensing data from such SNs and process them in time to help humans more efficiently. Sensor nodes (SNs) are processed in a timely manner to assist humans better in making decisions and responding to monitoring scenarios [4].

In this paper, we investigate the issue of the motion planning for UAV collecting low-power ground sensor node (SN) data in a dynamic jamming environment based on the following observation. In the past work, UAV-aided data collection scenarios were usually limited to simple path planning, which did not fully consider the principle of motion. Moreover, most of the past works are static environments or consider one-time offline global planning, and there is little research on dynamic environments that require real-time processing. Therefore, the research in this paper will build on past research to fully

consider the motion constraints of UAVs and real-time planning in dynamic environments. We use the mobility and flexibility of UAVs for SN data collection. When the UAV flies into the effective communication range of the SN (satisfying the demodulation threshold), it is activated to transmit data within the effective communication range. To achieve full autonomy in UAV data collection, we have taken several aspects into account in the design. Firstly, the motion planning plays an essential role in generating safe and smooth motions which consider motion primitives, flight energy consumption, and flight time during the data collection mission [5]. Secondly, to ensure that the data can be collected to meet the demodulation threshold, it is also necessary to consider the effective communication area [6].

*1.2. Prior Work*

We focus on past work from two fields: motion planning and data collection. To generate smooth and safe trajectories for UAVs online, the authors in [7,8] presented the extraction of free space in the configuration space and utilized a range of convex shapes to represent the free space. The motion planning problem is formulated as restricting motion trajectories to convex graphs through convex optimization, which can be solved directly with a dedicated toolbox. However, the size of the area in which the convex shapes are constructed directly affects the local optimality of the trajectory. The authors in [9] proposed a zoning method that can be combined with a multi-stage optimal control formulation to accommodate complex forms of unobstructed areas in unstructured environments due to the presence of multiple obstacles within the predicted range. This method can effectively increase the free space for UAV planning. Other authors have proposed methods that are simpler and faster to calculate, such as the geometry-based method in [10], which constructs convex polyhedra by means of ellipsoidal expansion.

Moreover, to constrain effectively the entire trajectory within the convex shape together with its derivatives in the feasible space under the hard constraint, the authors in [11] employed a piecewise Bézier curve on the basis of Bernoulli's polynomial to express the flight trajectory of the UAV. However, the order of the Bézier curve increases with the increase of control points, and a higher order can easily result in an "ill-conditioned" trajectory. To solve the problem of pathological Bézier curves under higher-order polynomials, the authors in [12] introduced B-spline curves, which have the advantages of Bézier curves while still being applicable to higher-order polynomial curves.

Collecting data from ground-based distributed SNs is among the crucial technologies for WSNs. The typical objectives of trajectory optimization are to minimize the mission time of the UAV [13–15], minimize energy consumption during the flight [16–18], maximize the amount of collected data [19,20], and maximize the amount of served sensors [21]. In addition to the basic UAV motion constraint, the data volume constraint is essential in all published research studies of UAV data collection. It is usually a non-convex constraint. In some studies, the communication rate required to meet the demodulation requirements is further considered, which is also non-convex. To deal with these non-convex optimization problems, design equivalence problems or relaxation problems as well as the transformation of constraints into penalty terms into optimization objectives have commonly been used in research. The successive convex approximation (SCA) [21] and some heuristic algorithms (e.g., PSO [22] and GE [23]) are commonly used to solve the trajectory. However, neither the SCA nor the heuristic algorithm can obtain a globally optimal solution, and the resulting local solution is strongly influenced by the initial value.

Inspired by the problems in past work, we focus on path planning and trajectory optimization considering UAV motion constraints in a dynamic jamming environment, which can be combined as motion planning. In our study, we first analyze the communication link [18,24] between the UAV and the ground sensors and formulate the optimization problem of minimizing the energy consumption, which considers the collection data demodulation threshold, obstacle avoidance, data collection volume, and motion principle. Then, we relaxed this non-convex optimization problem by introducing relaxation variables

and constructing equivalence constraints. Further, to solve the initial value problem of SCA during replanning in dynamic environments, we propose a communication-flight-corridor (CFC)-based initialized path generation method, which enables the SCA algorithm to converge to the optimal value quickly. Finally, the optimized path points were subjected to a B-spline curve.

### 1.3. Contributions and Organization

The goal of this study is the examination of UAV-aided data collection from deployed low-power ground SNs in dynamic jamming environments. The motion planning of the UAV is designed by considering the communication capability, motion primitives, and the low-power SNs when the sensors transmit data. The major contributions of this paper are as follows:

- An optimization framework was developed to minimize the energy consumption of UAV data collection tasks with communication link quality while maintaining constraints such as the maximum UAV speed, obstacle avoidance, and minimum data requirements per SNs.
- Based on the introduction of relaxation variables and the application of SCA, we rewrite the nonconvex constraints of the original problem and fit the discrete points through B-spline curves. To ensure that it is applicable to dynamic environments, we further devise a motion planning approach, which is dependent on reprogramming and updated in real time with a dynamic jamming environment.
- We present a CFC-based path initialization method, which enables the initial path to meet the constraints on the communication rate and the amount of collected data. In addition, we present a safe flight path correction method based on a geometric method for fast obstacle avoidance, which ensures that the initial path is safe.
- Simulation results validate the performance of the proposed algorithm under different parameter configurations and compare to the benchmark algorithms.

The remainder of this paper is structured as follows: In Section 2, we describe the investigated scenario and formulate the optimization problem through path discretization. Section 3 presents the designed non-convex solution for the problem. The simulation results in Section 4 validate the effectiveness of the presented algorithm. Lastly, the paper is summarized in Section 5.

### 2. Problem Statement and Formulation

In this study, a UAV is dispatched to collect data from a low-power SN on the ground. The data collection is performed with a time-division multiple-access protocol. Figure 1 shows the scenario: The UAV is flying at low altitude in a smart city. The wireless communication system comprises $K$ SNs represented by the set $\mathcal{K} = \{1, \ldots, K\}$, and the location of the SN $k$ is represented by $p_k = (x_k, y_k, 0), k \in \mathcal{K}$. The transmitted power for each of the low-power SNs is fixed at $P_k$. We consider the position of the SNs fixed and known for the trajectory design of the UAV. In practice, the position of the SNs can be obtained from the system database or identified with standard positioning techniques, such as with BeiDou systems or GPS [25]. In our scenario, the UAV arrives in the effective range of communication of the SN and activates the sensors for data transmission. It should be noted that the flight altitude of the UAV depends on the environment. Hence, it is simultaneously subject to air control by the local authorities. The complete flight altitude range is $H \in [H_{\min}, H_{\max}]$. The completion time of the data collection task is $T$, while the 3D trajectory of the UAV is denoted by $p(t) = (x(t), y(t), H(t))$, with $0 \leq t \leq T$. The completion time $T$ is discretized into $N$ sufficiently small and equal time slots with time discretization in which $\delta$ represents the length of the slot and $T = N\delta$. We introduce $\mathcal{N} = \{1, 2, \cdots, N\}$, in which $\delta$ is a variable. Consequently, the trajectory of the UAV in the 3D space in the $n$-th slot is expressed as follows: $p[n] = [x(n), y(n), H(n)], n = 1, 2, \ldots N + 1$. In addition, unlike in prior studies, there are $\mathcal{M} = \{1, \ldots, M\}$ no-fly zones that are blocked by obstacles or interference sources. These are represented by the set, and the location of the projected

center of the no-fly zone $m$ is $p_m = (x_m, y_m, 0), m \in \mathcal{M}$. For descriptive purposes, we refer to the no-fly zones as "obstacles" in what follows. Furthermore, there are $J$ jammers in the environment, which are represented by the set $\mathcal{J} = \{1, 2, \cdots, J\}$, and the position of the $j$-th jammer is $p_j[n] = (x_j(n), y_j(n), H_j(n)), j \in \mathcal{J}$. We assume that the position of the jammers is dynamic at a fixed height, which can obtain the location technology according to the radiation source [26,27].

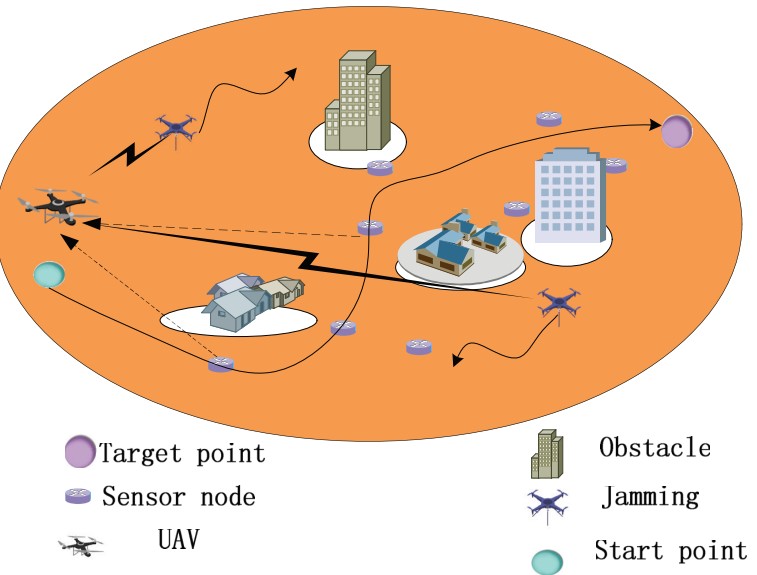

**Figure 1.** UAV-aided Data Collection with Dynamic Jamming.

## 2.1. Channel Model

The communication channel of the UAV has a crucial influence on data collection. Thus, we present the channel model first. The distance between the UAV and jammer $j$ in the $n$-th slot is as follows:

$$d_j[n] = \|p[n] - p_j[n]\|, \forall n \in \mathcal{N},\tag{1}$$

The distance between the UAV and SN $k$ in the $n$-th slot is as follows:

$$d_k[n] = \|p[n] - p_k\|, \forall n \in \mathcal{N}, k \in \mathcal{K},\tag{2}$$

We assume that the G2A channel comprises large- and small-scale fading. Small-scale fading is regarded as an identically distributed and independent Rician channel, and the Rice factor is $\mathcal{A}$; the channel coefficient $h_i[n]$ can be expressed as follows:

$$h_i[n] = \hat{h}_i \cdot g_i[n],\tag{3}$$

where $\hat{h}_i$ and $g_i[n]$ are the path-loss coefficients and small-scale fading, respectively. We can express the path-loss coefficient as follows:

$$g_i[n] = \sqrt{\mu_0 \cdot (d_i[n])^{-\alpha}},\tag{4}$$

where $\mu_0$ is the mean channel power gain over the reference distance $d_0 = 1$ m, and $\alpha$ is the path-loss exponent, which typically exceeds 2 for a Rician fading channel. Small-scale fading $\hat{h}_i$ consists of the LoS component $\bar{h}_i$, where $|\bar{h}_i| = 1$, and a random NLoS component $\tilde{h}_i$, where $\tilde{h}_i \sim \mathcal{CN}(0, 1)$. Small-scale fading $\hat{h}_i$ is determined as follows:

$$\hat{h}_i = \sqrt{\frac{\mathcal{A}}{\mathcal{A}+1}} \cdot \bar{h}_i + \sqrt{\frac{1}{\mathcal{A}+1}} \cdot \tilde{h}_i,\tag{5}$$

Assuming that each jammer is transmitting at a constant power $P_j$, the instantaneous signal to interference plus noise ratio (SINR) of the SN k is as follows:

$$\xi_k[n] = \frac{P_k x_k \mu_0 d_k[n]^{-\alpha}}{\sum\limits_{j=0}^{J-1} x_j \mu_0 d_j[n]^{-\alpha} + \sigma^2},\tag{6}$$

where $x_i = \left|\hat{h}_i\right|^2$, which is a random variable that obeys a non-central $\chi^2$ distribution. To simplify the optimization problem presented below, we further perform the homogeneous approximation of the SINR function, which considers the expected value of the SINR and yields the following result:

$$\mathbb{E}[\xi_k[n]] = \frac{P_k \cdot \beta_k \cdot d_k[n]^{-\alpha}}{\sum\limits_{j=0}^{J-1} P_j \cdot \beta_j \cdot d_j[n]^{-\alpha} + \sigma^2},\tag{7}$$

where

$$\beta_k = \int_0^\infty x_k \cdot f(x_k) \cdot \mu_0 \cdot dx_k,\tag{8}$$

$$\beta_j = \int_{\substack{0 \\ x_0,\cdots x_{J-1}}}^\infty \cdots \int_0^\infty x_j \mu_0 \prod_j^{J-1} f(x_j) dx_0 \cdots dx_{J-1},\tag{9}$$

$$f(x) = \frac{\mathcal{A}+1}{\mathcal{P}} \exp(-\mathcal{A} - \frac{(\mathcal{A}+1)x}{\mathcal{P}}) I_0(2\sqrt{\frac{\mathcal{A}(\mathcal{A}+1)x}{\mathcal{P}}}),\tag{10}$$

where $\mathcal{P}$ is the signal model parameter, which, in general, can be set to 1. Moreover, $I_0(\cdot)$ is a Bessel function of order zero. $\beta_k$ and $\beta_j$ are constants in each independent simulation that can be generated according to the Rician channel model. When the time slot is sufficiently small or when the flight speed is low, we consider the position of the UAV a fixed point in each time slot. The expected information rate (bits/s) between the UAV and SN $k$ in the $n$-th slot is represented as follows:

$$r_k[n] = \frac{B}{K} \cdot \log_2(1 + \frac{P_k \cdot \beta_k \cdot d_k[n]^{-\alpha}}{\sum\limits_{j=0}^{J-1} P_j \cdot \beta_j \cdot d_j[n]^{-\alpha} + \sigma^2}).\tag{11}$$

where $B$ is the total channel bandwidth in Hz, and $\sigma^2$ represents the noise power spectral density.

### 2.2. Motion Primitives

The set $s[n] \in \mathcal{S}$ is a dynamical system composed of the 3-*D* position, acceleration, and velocity: $s[n] = [p[n], v[n], a[n]]^{\mathrm{T}}$. According to [28], the differential flatness of the UAV allows us to build control inputs from a 1-*D* time-parameterized polynomial trajectory that is independently specified on each axis. The discrete state over the entire task period can be $\mathbf{P} = [p[1], p[2], \cdots, p[N]]^{\mathrm{T}}$, $\mathbf{V} = [v[1], v[2], \cdots, v[N]]^{\mathrm{T}}$, and $\mathbf{A} = [a[1], a[2], \cdots, a[N]]^{\mathrm{T}}$. The

control variable input for the system is the jerk: $\mathbf{J} = [j[0], j[1], \cdots, j[N-1]]^{\mathrm{T}}$. According to the equation of motion, the following correlations can be acquired:

$$p[n+1] = p[n] + v[n] \cdot \delta + \frac{1}{2} \cdot a[n] \cdot \delta^2$$
$$+ \frac{1}{6} \cdot j[n] \cdot \delta^3, n = 0, \cdots, N-1 \tag{12}$$

$$v[n+1] = v[n] + a[n] \cdot \delta$$
$$+ \frac{1}{2} \cdot j[n] \cdot \delta^2, n = 0, \cdots, N-1 \tag{13}$$

$$a[n+1] = a[n] + j[n] \cdot \delta, n = 0, \cdots, N-1 \tag{14}$$

The matrix forms are as follows:

$$\mathbf{P} = \mathbf{T}_p \cdot \mathbf{J} + \mathbf{B}_p, \tag{15}$$

$$\mathbf{V} = \mathbf{T}_v \cdot \mathbf{J} + \mathbf{B}_v, \tag{16}$$

$$\mathbf{A} = \mathbf{T}_a \cdot \mathbf{J} + \mathbf{B}_a, \tag{17}$$

where

$$\mathbf{T}_p = \begin{bmatrix} \frac{1}{6}\delta^3 & 0 & 0 & \cdots & 0 \\ \frac{7}{6}\delta^3 & \frac{1}{6}\delta^3 & 0 & \cdots & 0 \\ \vdots & \vdots & \vdots & \cdots & 0 \\ \frac{3N+1}{6}\delta^3 & \cdots & \frac{3(N-n+1)\cdot(N-n)+1}{6}\delta^3 & \cdots & \frac{1}{6}\delta^3 \end{bmatrix}_{N \times N}, \tag{18}$$

$$\mathbf{B}_p = \begin{bmatrix} p[0] + v[0]\delta + \frac{1}{2}a[0]\delta^2 \\ p[0] + 2v[0]\delta + \frac{2^2}{2}a[0]\delta^2 \\ \vdots \\ p[0] + Nv[0]\delta + \frac{N^2}{2}a[0]\delta^2 \end{bmatrix}_{N \times 1}, \tag{19}$$

$$\mathbf{T}_v = \begin{bmatrix} \frac{1}{2}\delta^2 & 0 & 0 & \cdots & 0 \\ \frac{3}{2}\delta^2 & \frac{1}{2}\delta^2 & 0 & \cdots & 0 \\ \vdots & \vdots & \vdots & \cdots & 0 \\ (N-\frac{1}{2})\delta^2 & \cdots & (N-n+\frac{1}{2})\delta^2 & \cdots & \frac{1}{2}\delta^2 \end{bmatrix}_{N \times N}, \tag{20}$$

$$\mathbf{B}_v = \begin{bmatrix} v[0] + a[0]\delta \\ v[0] + 2a[0]\delta \\ \vdots \\ v[0] + Na[0]\delta \end{bmatrix}_{N \times 1}, \tag{21}$$

$$\mathbf{T}_a = \begin{bmatrix} \delta & 0 & 0 & \cdots & 0 \\ \delta & \delta & 0 & \cdots & 0 \\ \vdots & \vdots & \vdots & \cdots & 0 \\ \delta & \delta & \delta & \cdots & \delta \end{bmatrix}_{N \times N} , \tag{22}$$

$$\mathbf{B}_a = \begin{bmatrix} a[0] \\ a[0] \\ \vdots \\ a[0] \end{bmatrix}_{N \times 1} . \tag{23}$$

in which $p[0]$ is the initial position, $v[0]$ is the initial velocity, and $a[0]$ is the initial acceleration.

*2.3. Problem Formulation*

The optimization goal is to minimize the energy consumption and flight time of the UAVs when performed data collection missions. The optimization goal is to keep the flight time and energy consumption of the UAV minimal while collecting data. The energy consumption of a UAV typically comprises two major components: the propulsion energy and communication-related energy [18]. The latter consists of the energy utilized for communication circuits, signal radiation/reception, signal processing, and so on. For the purpose of this study, we consider the communication-related energy constant. To maintain the UAV at high altitude and to support its motion, propulsion energy is consumed. Typically, the propulsion energy depends on the flight speed, acceleration, jerk of the UAV, and so on. We take interest in the effort or smoothness of the trajectory, i.e., the square $L^2$-norm of the control input $j[n]$, $a[n]$ and $v[n]$, which represent the energy cost in a dynamic system. Hence, we express the energy optimization function as below:

$$\begin{aligned} f_1(\mathbf{J}) &= \omega_1 \mathbf{V}^T \mathbf{V} + \omega_2 \mathbf{A}^T \mathbf{A} + \omega_3 \mathbf{J}^T \mathbf{J} \\ &= \mathbf{J}^T \left( \omega_1 \mathbf{T}_v^T \mathbf{T}_v + \omega_2 \mathbf{T}_a^T \mathbf{T}_a + \omega_3 \mathbf{I} \right) \mathbf{J} + \\ &\quad 2 \left( \omega_1 \mathbf{B}_v^T \mathbf{T}_v + \omega_2 \mathbf{B}_a^T \mathbf{T}_a \right) \mathbf{J} + constant. \end{aligned} \tag{24}$$

where $f_1(\mathbf{J})$ represents the smoothness and energy consumption of the UAV trajectory. $\mathbf{I}$ is the identity matrix; $\omega_1$, $\omega_2$, $\omega_3$ are the weighting factors. The $constant = \omega_1 \mathbf{B}_v^T \mathbf{B}_v + \omega_2 \mathbf{B}_a^T \mathbf{B}_a$ does not need to be considered in the optimization. Therefore, we formulate the optimization problem as follows:

$$\begin{aligned} \text{P0}: \quad \min_{J,N} \quad & \mathbf{J}^T \left( \omega_1 \mathbf{T}_v^T \mathbf{T}_v + \omega_2 \mathbf{T}_a^T \mathbf{T}_a + \omega_3 \mathbf{I} \right) \mathbf{J} + \\ & 2 \cdot \left( \omega_1 \mathbf{B}_v^T \mathbf{T}_v + \omega_2 \mathbf{B}_a^T \mathbf{B}_a \right) \mathbf{J} \end{aligned} \tag{25}$$

$$st: \quad \mathbf{P} = \mathbf{T}_p\mathbf{J} + \mathbf{B}_p \tag{26a}$$

$$\mathbf{V} = \mathbf{T}_v\mathbf{J} + \mathbf{B}_v \tag{26b}$$

$$\mathbf{A} = \mathbf{T}_a\mathbf{J} + \mathbf{B}_a \tag{26c}$$

$$-\mathbf{V}_{\max} - \mathbf{B}_v \leqslant \mathbf{T}_v\mathbf{J} \leqslant \mathbf{V}_{\max} - \mathbf{B}_v \tag{26d}$$

$$-\mathbf{A}_{\max} - \mathbf{B}_a \leqslant \mathbf{T}_a\mathbf{J} \leqslant \mathbf{A}_{\max} - \mathbf{B}_a \tag{26e}$$

$$\mathbf{H}_{\min} - \mathbf{B}_{p_z} \leqslant \mathbf{T}_p\mathbf{J}_z \leqslant \mathbf{H}_{\max} - \mathbf{B}_{p_z} \tag{26f}$$

$$p[0] = p_0, v[0] = v_0, a[0] = a_0 \tag{26g}$$

$$p[N] = p_N, v[N] = v_N, a[N] = a_N \tag{26h}$$

$$\sum_{n=1}^{N} \zeta_n^k r_k[n]\delta \geqslant Q_{th}^k, \forall k \in \mathcal{K} \tag{26i}$$

$$\zeta_n^k = \mathcal{I}\{\max(r_k[n] - r_{th}, 0)\}, \forall n \in \mathcal{N} \tag{26j}$$

$$\mathbf{P} \notin \mathbf{obstacles} \tag{26k}$$

and

$$\mathcal{I}(x) = \begin{cases} 1, x \geqslant 0 \\ 0, x \leqslant 0 \end{cases} \tag{27}$$

where (26a), (26b) and (26c) are the UAV motion primitives based on model prediction. (26d) and (26e) are the motion constraints of the UAV, which limit the range of the maximum velocity and maximum acceleration: and $\mathbf{V}_{\max} = v_{\max} \cdot \mathbf{I}_{N \times 1}$ and $\mathbf{A}_{\max} = a_{\max} \cdot \mathbf{I}_{N \times 1}$. (26f) is the height constraint of the UAV, and (26g) and (26h) are the start and end state constraints. (26i) represents the collected data amount of the $k$-th sensor, where $\zeta_n^k$ is the effective collection indication, and $Q_{th}^k$ is the minimum collection requirement. (26j) is the minimum communication rate of the sensor given according to the actual communication system. Moreover, (26k) is the obstacle avoidance constraint, which is usually a non-convex constraint that limits the UAV motions to a safe area. The formulated problem P0 is difficult to solve directly for the following reasons: (i) The mission completion time of UAVs is closely related to the dimensions of other state variables, which make it difficult to determine the constraints; (ii) the constraints (26i) and (26k) are non-convex. To this end, we solve this problem through the introduction of slack variables and the use of SCA.

## 3. Global Replanning Based on CFC

### 3.1. General Framework

The major challenge in optimally solving (P0) is to optimize the variable N and non-convex constraints (26i) and (26k), which involve uncertainty of other optimization variables and indicator functions (26j) with respect to the UAV trajectory. Without affecting the optimality of (P0), the trajectory of the UAV can be considered to constitute only the connecting line segment [6]. This conclusion means that finding the optimal solution to (P0) is comparable to finding an ordered set of waypoints containing positions that represent the start and end points of each segment and to optimizing the instantaneous speed of the UAV on the trajectory linking these waypoints. To this end, as shown in Figure 2, the task time N and trajectory energy consumption are decoupled in the optimization objective. First, minimum time discrete path initialization based on a CFC is proposed under the condition that the minimum discrete-time slot is $\delta$. Second, we use the SCA algorithm to optimize the global path according to the initialization path. Third, a B-spline method is used to smooth the trajectory in a minimum discrete-time slot, $\delta$. After executing the trajectory of the time slot $\delta$, the communication channel state and state information of the UAV itself (including the position, velocity, and acceleration) are re-estimated and updated. Finally, the operation starting from path initialization is repeated until the destination point is reached. We will

present the path initialization, path optimization, and B-spline trajectory optimization in more detail.

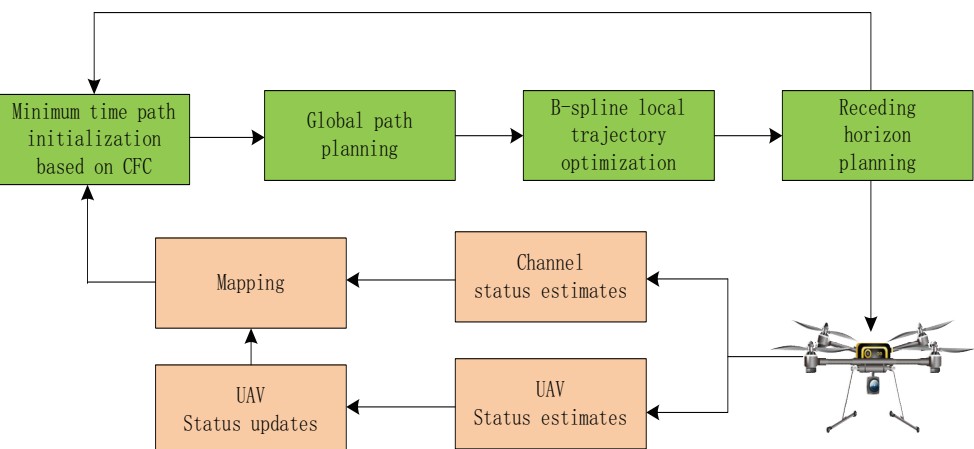

**Figure 2.** Block diagram of presented autonomous system.

### 3.2. Global Path Planning Based on SCA

For the nonconvex problem $P0$, we perform the following for the nonconvex constraints (26i), constraints (26j) and constraints (26k). We first introduce the auxiliary variables $\kappa[n]$ for constraints (26i) and constraints (26j) in the following equivalent form: For the non-convex problem P0, we perform the following steps for the non-convex constraints (26j) and constraints (26j) and constraints (26k). First, we introduce the auxiliary variables $\kappa[n]$ for the constraints (26i) and (26j) with the following equivalent form:

$$\sum_{n=1}^{N} \kappa_k[n] \cdot r_k[n] \cdot \delta \geqslant Q_{th}, \forall k \in \mathcal{K}, \tag{28}$$

$$\kappa_k[n] \cdot r_k[n] \geqslant \kappa_k[n] \cdot r_{th}, \forall k \in \mathcal{K}, \tag{29}$$

$$\kappa_k[n] \in \{0,1\}, \forall k \in \mathcal{K}. \tag{30}$$

**Proof.** Constraint (28) shows that $\kappa_k[n]$ can only be 1 when $r_k[n] \geqslant r_{th}$. Otherwise, it is zero. The minimum amount of data collected constantly must be higher than $Q_{th}$, and the constraint (28) is only valid when some $\kappa_k[n]$ equal to 1. Hence, the constraints (28) and (29) can make the UAV "pass through" the "surrounding" of each SN during its flight while remaining at a low velocity for a period of time based on the minimum amount of collected data. □

However, constraints (28) and (29) are still non-convex. For $r_k[n]$, the auxiliary variables $S_k[n]$ and $I_k[n]$ are introduced to obtain the following relation:

$$\sum_{n=1}^{N} \kappa_k[n] \cdot \tilde{r}_k[n] \cdot \delta \geqslant Q_{th}, \tag{31}$$

where $r_k[n] \geqslant \tilde{r}_k[n] = B \cdot \log_2(1 + \frac{S_k[n]^{-1}}{I_k[n]})$. The auxiliary variables $S_k[n]$ and $I_k[n]$ must meet the following conditions:

$$S_k[n]^{-1} \leqslant \beta_0 p_0 d_k[n]^{-\alpha}, S_k[n] > 0, \tag{32}$$

$$I_k[n] \geqslant \sum_j \beta_j p_j d_j[n]^{-\alpha} + \sigma^2. \tag{33}$$

For constraint (32), we can rewrite the convex constraint as follows:

$$d_k[n]^\alpha - \beta_0 p_0 S_k[n] \leqslant 0, S_k[n] > 0. \tag{34}$$

However, constraint (33) is still non-convex. We perform a first-order Taylor expansion:

$$\begin{aligned}
d_j^{-\alpha}[n] &= \left[ (p_x - p_{jx})^2 + (p_y - p_{jy})^2 + (p_z - p_{jz})^2 \right]^{-\alpha/2} \\
&\geqslant L_j^r[n]^{-\alpha/2} - \alpha L_j^r[n]^{-\alpha/2-1} \left[ (p^r - p_j)^T \cdot (p - p^r) \right],
\end{aligned} \tag{35}$$

where $L_j[n] = (p_x - p_{jx})^2 + (p_y - p_{jy})^2 + (p_z - p_{jz})^2$, and $L_j^r[n]$, $p^r$ are the initial values of the *r*-th iteration. This results in a new convex constraint (36), which is a relaxation of constraint (33):

$$I_k[n] \geqslant \sum_j \beta_j p_j \left( L_j^r[n]^{-\alpha/2} - A_j^r[n] \right) + \sigma^2, \tag{36}$$

where $A_j^r[n] = \alpha L_j^r[n]^{-\alpha/2-1} \left[ (p^r - p_j)^T \cdot (p - p^r) \right]$. For constraint (31), we further introduce an auxiliary variable $\varepsilon[n]$, that is less than the first-order Taylor expansion of $\tilde{r}_k[n]$.

$$\begin{aligned}
\varepsilon_k[n] &\leqslant U_k^r[n] + V_k^r[n](S_k[n] - S_k^r[n]) \\
&+ W_k^r[n](I_k[n] - I_k^r[n]),
\end{aligned} \tag{37}$$

where $U_k^r[n] = B \cdot \log_2(1 + \frac{1}{S_k^r[n] I_k^r[n]})$, $V_k^r[n] = \frac{-B \cdot \log_2 e}{S_k^r[n] + S_k^r[n]^2 I_k^r[n]}$, and $W_k^r[n] = \frac{-B \cdot \log_2 e}{I_k^r[n] + I_k^r[n]^2 S_k^r[n]}$. Therefore, a further relaxed form of the constraint (31) can be obtained

$$\sum_{n=1}^N \kappa_k[n] \cdot \varepsilon_k[n] \cdot \delta \geqslant Q_{th}, \tag{38}$$

Constraint (38) is still a non-convex constraint owing to the product of the indicator function variable $\kappa_k[n]$ and variable $\varepsilon_k[n]$. Thus, we continue to relax the indicator function:

$$0 \leqslant \kappa_k[n] \leqslant 1, \tag{39}$$

$$\kappa_k[n] \cdot (\kappa_k[n] - 1) \geqslant 0. \tag{40}$$

The convex constraints can be obtained via a Taylor expansion and $\kappa_k[n] \cdot \varepsilon_k[n] = \frac{(\kappa_k[n] + \varepsilon_k[n])^2 - (\kappa_k[n] - \varepsilon_k[n])^2}{4}$ for constraints (29), (38) and (40)

$$\begin{aligned}
&H_k^r[n]^2 - 2H_k^r[n](\kappa_k[n] + \varepsilon_k[n]) \\
&+ (\kappa_k[n] - \varepsilon_k[n])^2 \geqslant 4\kappa_k[n] \cdot r_{th},
\end{aligned} \tag{41}$$

$$\begin{aligned}
\sum_{n=1}^N &[H_k^r[n]^2 - 2H_k^r[n] \cdot (\kappa_k[n] + \varepsilon_k[n]) \\
&+ (\kappa_k[n] - \varepsilon_k[n])^2] \cdot \delta \geqslant 4Q_{th},
\end{aligned} \tag{42}$$

$$2\kappa_k^r[n] \cdot \kappa_k[n] - \kappa_k[n] - \kappa_k^r[n]^2 \geqslant 0. \tag{43}$$

where $H_k^r[n] = (\kappa_k^r[n] + \varepsilon_k^r[n])$. Thus, we obtain the new convex constraints (41), (42) and (43) of problem P1 after the equivalent transformation and relaxation of constraints (26i) and (26j). In the next step, we continue to deal with the non-convex constraint (26k) of problem P1. The obstacles in the environment are modeled as cylinders. The UAV must keep a certain safe distance to them during its flight. We set the safe distance as $d_{safe}$. During the task, the UAV should ensure that the distance between the projection coordinates and the center coordinates of the bottom surface of the cylindrical obstacle exceeds the safe distance or that its flight height is much higher than that of the cylinder to ensure a safe flight. The respective safety constraint between the UAV and obstacle $m$ is as follows:

$$\begin{aligned} \left\| p_{xy}[n] - p_m \right\| &\geqslant d_{safe} \\ or \quad p_z[n] &\geqslant h_m + d_{safe}. \end{aligned} \tag{44}$$

This is an "or" constraint that is non-convex. In addition, $p_{xy}[n]$ represents the two-dimensional coordinates projected by the UAV onto the ground, and $p_z[n]$ is the flight height of the UAV. By introducing the variables 0 and 1, we can express the equivalent form of the constraint (44) as follows:

$$\left\| p_{xy}[n] - p_m \right\| + \lambda_{xy,m}[n] \cdot C \geqslant d_{safe}, \tag{45}$$

$$p_z[n] + \lambda_{z,m}[n] \cdot C \geqslant h_m + d_{safe}, \tag{46}$$

$$\lambda_{xy,m}[n] + \lambda_{z,m}[n] \geqslant 1, \tag{47}$$

$$\lambda_{xy,m}[n], \lambda_{z,m}[n] \in \{0, 1\}. \tag{48}$$

where $C$ is a big positive constant and $\lambda_{xy,m}[n], \lambda_{z,m}[n]$ are is a binary number. We further relax these binary integer constraints

$$\begin{aligned} \frac{(p_{xy}^r[n] - p_m)^T \cdot (p_{xy}[n] - p_{xy}^r[n])}{A_m^r[n]} &\geqslant \\ d_{safe} - \lambda_{xy,m}[n] \cdot C - A_m^r[n]. \end{aligned} \tag{49}$$

where $A_m^r[n] = \left\| p_{xy}^r[n] - p_m \right\|$. By relaxing $\lambda_{xy,m}$ and $\lambda_{z,m}$ in constraint (48), the new constraints can be obtained as follows:

$$0 \leqslant \lambda_{i,m}[n] \leqslant 1, i \in \{xy, z\}, \tag{50}$$

$$\lambda_{i,m}[n](\lambda_{i,m}[n] - 1) \geqslant 0, i \in \{xy, z\}, \tag{51}$$

The relaxation form of the non-convex constraint (51) can be obtained via a Taylor expansion

$$-\lambda_{i,m}^r[n]^2 + 2\lambda_{i,m}^r[n]\lambda_{i,m}[n] - \lambda_{i,m}[n] \geqslant 0. \tag{52}$$

where $\lambda_{i,m}^r[n]$ is the $r$-th iteration of $\lambda_{i,m}[n]$. So far, we have dealt with all the non-convex constraints in P0 to obtain the new optimization problem as follows:

$$\begin{aligned} \text{P1}: \quad &\min_{J,\kappa,\lambda,\varepsilon,I,S} \mathbf{J}^T \left( \omega_1 \mathbf{T}_v^T \mathbf{T}_v + \omega_2 \mathbf{T}_a^T \mathbf{T}_a + \omega_3 \mathbf{I} \right) \mathbf{J} + \\ &2 \cdot \left( \omega_1 \mathbf{B}_v^T \mathbf{T}_v + \omega_2 \mathbf{B}_a^T \mathbf{B}_a \right) \mathbf{J} \end{aligned} \tag{53}$$

$$st : \quad \mathbf{P} = \mathbf{T}_p \mathbf{J} + \mathbf{B}_p \tag{54a}$$

$$\mathbf{V} = \mathbf{T}_v \mathbf{J} + \mathbf{B}_v \tag{54b}$$

$$\mathbf{A} = \mathbf{T}_a \mathbf{J} + \mathbf{B}_a \tag{54c}$$

$$- \mathbf{V}_{\max} - \mathbf{B}_v \leqslant \mathbf{T}_v \mathbf{J} \leqslant \mathbf{V}_{\max} - \mathbf{B}_v \tag{54d}$$

$$- \mathbf{A}_{\max} - \mathbf{B}_a \leqslant \mathbf{T}_a \mathbf{J} \leqslant \mathbf{A}_{\max} - \mathbf{B}_a \tag{54e}$$

$$\mathbf{H}_{\min} - \mathbf{B}_{p_z} \leqslant \mathbf{T}_p \mathbf{J}_z \leqslant \mathbf{H}_{\max} - \mathbf{B}_{p_z} \tag{54f}$$

$$p[0] = p_0, v[0] = v_0, a[0] = a_0 \tag{54g}$$

$$p[N] = p_N, v[N] = v_N, a[N] = a_N \tag{54h}$$

$$\begin{aligned} H_k^r[n]^2 - 2H_k^r[n](\kappa_k[n] + \varepsilon_k[n]) \\ + (\kappa_k[n] - \varepsilon_k[n])^2 \geqslant 4\kappa_k[n] \cdot r_{th} \end{aligned} \tag{54i}$$

$$\begin{aligned} \sum_{i=1}^{N} [H_k^r[n]^2 - 2H_k^r[n] \cdot (\kappa_k[n] + \varepsilon_k[n]) \\ + (\kappa_k[n] - \varepsilon_k[n])^2] \cdot \delta \geqslant 4Q_{th} \end{aligned} \tag{54j}$$

$$2\kappa_k^r[n] \cdot \kappa_k[n] - \kappa_k[n] - \kappa_k^r[n]^2 \geqslant 0 \tag{54k}$$

$$\begin{aligned} \frac{(p_{xy}^r[n] - p_m)^T \cdot (p_{xy}[n] - p_{xy}^r[n])}{A_m^r[n]} \geqslant \\ d_{safe} - \lambda_{xy,m}[n] \cdot C - A_m^r[n] \end{aligned} \tag{54l}$$

$$p_z[n] + \lambda_{z,m}[n] \cdot C \geqslant h_m + d_{safe} \tag{54m}$$

$$\lambda_{xy,m}[n] + \lambda_{z,m}[n] \geqslant 1 \tag{54n}$$

$$0 \leqslant \lambda_{i,m}[n] \leqslant 1, i \in \{xy, z\} \tag{54o}$$

$$- \lambda_{i,m}^r[n]^2 + 2\lambda_{i,m}^r[n]\lambda_{i,m}[n] - \lambda_{i,m}[n] \geqslant 0 \tag{54p}$$

Evidently, the problem is a convex optimization problem that can be addressed through the application of standard convex optimization technology or CVX (for instance, the interior-point method). However, some of the constraints in the previously presented optimization problem are sensitive to the initial value after relaxation. Although we have designed a continuous convex approximation method to solve the problem, the influence of the initial value is still not negligible. It will influence the convergence time of the algorithm and the optimal solution. To this end, we designed a CFC-based initialization method, which will be described in detail in the next section.

*3.3. Minimum Time Path Initialization Based on CFC*

In practice, the communication capability of SNs is limited by the power and environment. It is usually necessary to calculate the communication link under the constraints of the demodulator threshold, bit error rate, receiver sensitivity, and so on to determine the supported communication range. We propose flight path segmentation based on a CFC. Full-speed flight paths are generated according to the trapezoidal criterion [5]. To find the CFC, we define the effective communication area of the SN $k$ free of interferers: $I_{j,k} = \min\{\beta_0 p_j d_{k,j}^{-\alpha}, \beta_0 p_j d_{\max}^{-\alpha}\}$, where is the trajectory within the effective communication area of the SN, and is the maximum distance. To produce a convex CFC from the jammers and SN, the following procedures are performed:

1.  *Find the nearest jammer*: calculate the distance between all the jammers and SN $k$, choose the nearest jammer $j^*$, and let the jamming value of the other jammers be $I_{j,k} = \min\{\beta_0 p_j d_{k,j}^{-\alpha}, \beta_0 p_j d_{\max}^{-\alpha}\}$, which is the investigated worst-case scenario we have in mind because the jamming value of interferer $j$ would not surpass $I_{j,k}$ in the SN $k$ communication region.

2. *Find critical points*: on the line segment between jammer $j^*$ and the central position of SN $k$, find a point $P_{j^*,k}$ along the direction of the center of the SN communication circle that will meet the communication rate larger than $R_{\min}$.

3. *Find CFC*: pass through $p_{j^*,k}$ and ensure that the perpendicular $l_{k,j^*}$ is on a line between jammer $j^*$ and SN $k$.

The intersection of the half-space on the vertical $l_{k,j^*}$ side of SN $k$ and the communication range of SN $k$, is the CFC, which is a convex polyhedron. The detailed construction process is described in Algorithm 1. It divides the UAV path into two paths: the communication path and full-speed flight path. The UAV data are collected on the communication path, which is within the effective communication range.

---

**Algorithm 1:** Given the position of SN $k$ and the position of jammer, find $CFC_k$.

---

1: **function** Find_CFC($p_k$, **p**, $p_j$)

2: Set the jammer set *jammer_open_list* $= \{1, 2, \cdots, J\}$ and *jammer_close_list* $= \{\varnothing\}$.

3: Search for jammer closest to the sensor node:

4:     $j^* = \arg \min_{j \in \mathcal{J}} \left\| p_k - p_j \right\|$

5: Remove $j^*$ from the set:

6:     *jammer_open_list* $\to j^*$

7:     *jammer_close_list* $\leftarrow j^*$

8: Set the energy of the remaining jammer:

9:     $I_{j,k} = \min\{\beta_0 P_j d_{k,j}^{-\alpha}, \beta_0 P_j d_{\max}^{-\alpha}\}$

10: Get the line segment $[p_k, p_{j^*}]$ between the nearest jammer, and search for point $p_{j^*,k}$ to make it meet:

    $p_{j^*,k} = \arg \max_{p_{m,k} \in [p_k, p_{j^*}]} \left\| p_{j,k} - p_k \right\|$

11:     $s.t. \quad B \cdot \log_2 \left(1 + \frac{\beta_0 p_k \|p[n] - w_k\|^{-\alpha}}{I_k + \sigma^2}\right) \geq R_{\min}$

    $I_k = \sum_{j=0, j \neq j^*}^{M} I_{j,k} + \beta_0 p_k \left\| p_{j,k} - p_k \right\|^{-\alpha}$

12: **if** there is no $p_{j^*,k}$, it is determined that SN $k$ cannot perform the data collection, stop algorithm.

13: **else** continue

14: Find the perpendicular of line segment $[p_k, p_{j^*}]$ through point $p_{j^*,k}$:

15:     $a_{j^*}^T w = b_{j^*}$

16: Search for jammer $j'$ belonging to $C_{k,n}(p_k)$ and perpendicular to line segment $[p_k, p_{j^*}]$ through $w_{m'}$:

17:     $a_{j'}^T w = b_{j'}$

18: combine perpendiculars:

19:     $\mathbf{A}_k^{\dagger} \leftarrow \begin{bmatrix} a_0^T \\ a_1^T \\ \vdots \end{bmatrix}, \mathbf{b}_k \leftarrow \begin{bmatrix} b_0 \\ b_1 \\ \vdots \end{bmatrix}$

20: Calculate the CFC:

21:     $\Omega_{k,n} = C_{k,n}(p_k) \cap \left\{\mathbf{A}_k^{\dagger} \cdot \mathbf{p} \leq \mathbf{b}_k\right\}, \Omega_j \leftarrow \Omega_{k,n}$

22: **end**

---

After acquiring the CFC, the initialization generates discrete points on the path according to the "trapezoidal principle" because we expect the UAV to be as slow as possible when approaching the SN during data collection. Thus, it needs enough time to complete the task. We conduct the analysis for the case of only one SN, as shown in Figure 3 (the green area is the CFC).

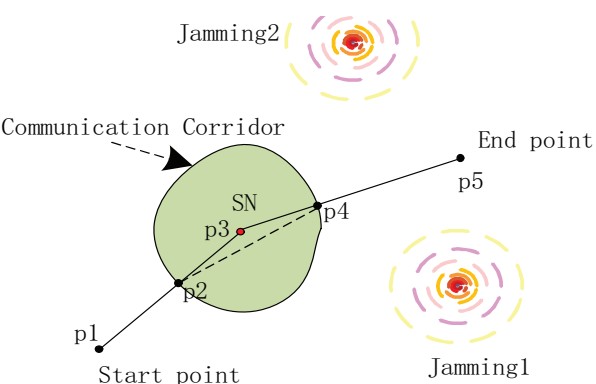

**Figure 3.** Path initialization based on CFC.

We first connect the starting point $p_1$ to sensor position point $p_3$ and terminal position point $p_5$. The intersection points of the line segment and CFC are denoted as $p_2$ and $p_4$. Furthermore, $|p_1 p_2|$ allocates the time and determines the position of discrete points according to the trapezoidal preparation. The UAV accelerates from the initial position to the maximum velocity, remains at the maximum velocity for a while and finally gradually slows down to zero velocity to reach the end point. The total flight time of this period is $t_1$. In addition, the acceleration period is $t_{11}$, the period of uniform movement is $t_{12}$, and the period of deceleration is $t_{13}$ such that $t_1 = t_{11} + t_{12} + t_{13}$, where $t_{11} = \frac{v_{\max}}{a_{\max}} = t_{13}$, $t_{12} = \frac{d_1}{v_{\max}} - \frac{v_{\max}}{a_{\max}}$, and $d_1$ is the distance of $|p_1 p_2|$. Therefore, the number of discrete points on the $|p_1 p_2|$ segment can be obtained as follows:

$$N_{11} = round(\frac{v_{\max}}{a_{\max} \cdot \delta}) = N_{13}, \tag{55}$$

$$N_{12} = round(\frac{d_1}{v_{\max} \cdot \delta} - \frac{v_{\max}}{a_{\max} \cdot \delta}). \tag{56}$$

Consequently, the discrete points of segment $|p_1 p_2|$ can be expressed as follows:

$$p_1(n) = \begin{cases} p_1(0) + \frac{1}{2} a_{\max} \cdot (n \cdot \delta)^2, & n = 0, \cdots, N_{11} - 1 \\ p_1(N_{11} - 1) + v_{\max} \cdot n \cdot \delta, & n = 1, \cdots, N_{12} \\ p_1(N_{12}) + v_{\max} \cdot n \cdot \delta - \frac{1}{2} a_{\max} \cdot (n \cdot \delta)^2, & n = 1, \cdots, N_{13} \end{cases} \tag{57}$$

$|p_2 p_3|$ and $|p_3 p_4|$ are located in the CFC and can meet the minimum communication rate. We suppose that the time $\frac{Q_{th}}{r_{th}}$ is necessary to collect data at the minimum rate $r_{th}$ and the requested minimum quantity $Q_{th}$. This time period comprises two flight paths with the flight times $t_2$ and $t_3$, respectively:

$$t_2 + t_3 = \frac{Q_{th}}{r_{th}}, \tag{58}$$

$$N_2 + N_3 = \frac{Q_{th}}{r_{th} \cdot \delta}. \tag{59}$$

We let these two trajectories move at a constant speed. Thus, $\frac{N_2}{N_3} = \frac{d_2}{d_3}$, where $d_2$ and $d_3$ are the distances of $|p_2 p_3|$ and $|p_3 p_4|$, respectively. We determine $N_2$ and $N_3$ as follows:

$$N_2 = \frac{Q_{th}}{r_{th} \cdot \delta} \cdot \frac{d_2}{d_2 + d_3}, \tag{60}$$



$$N_3 = \frac{Q_{th}}{r_{th} \cdot \delta} \cdot \frac{d_3}{d_2 + d_3}, \tag{61}$$

$$p_2(n) = (1 - \frac{n}{N_2}) \cdot p_2(0) + \frac{n}{N_2} \cdot p_2(N_2 - 1), n = 1, \cdots, N_2, \tag{62}$$

$$p_3(n) = (1 - \frac{n}{N_3}) \cdot p_3(0) + \frac{n}{N_3} \cdot p_3(N_3 - 1), n = 1, \cdots, N_3. \tag{63}$$

where $p_2(0) = p_2$ and $p_3(0) = p_3$. $|p_4 p_5|$ and $|p_1 p_2|$ can be processed in the same way. Therefore, we can obtain the initial waypoints of the global plan according to Algorithm 2.

---

**Algorithm 2:** For a given initial position $p_0$, end position $p_N$, and position of the jammer and SN, find the initial global discrete path.

1: **function** $Initial\_CFC\_path(p_0, p_N, p_{SN}, p_{jammer})$
2: Compute the *CFC* of the *SN* with Algorithm 1:
3:   Find_CFC($p_{SN}, p_{UAV}, p_{jammer}$)
4: Set the initial point to $p_1$, the end point to $p_5$, and the *SN* position to $p_3$:
5:   $p_2 = p_1 p_3 \cap CFC, p_4 = p_3 p_5 \cap CFC$
6: **if** UAV is not in CFC and $Q_{current} \leqslant Q_{th}$
7:   Calculate the discrete points of $|p_1 p_2|$ and $|p_4 p_5|$ according to Equation (57)
8:   Calculate the discrete points of $|p_2 p_3|$ and $|p_3 p_4|$ according to Equations (60) and (61)
9: **elsif** UAV is in CFC and $Q_{current} \leqslant Q_{th}$
10:   Calculate the discrete points of $|p_4 p_5|$ according to Equation (57)
11:   Calculate the discrete points of $|p_2 p_3|$ and $|p_3 p_4|$ according to Equations (60) and (61)
12: **else**
13:   Calculate the discrete points of $|p_{UAV} p_5|$ according to Equation (57)
14: Output the path discrete point:
15:   $\mathbf{p}_{initCFC} = \{p_1(0) \cdots p_2(0) \cdots p_3(0) \cdots p_4(0) \cdots \}$
16: **end**

---

It is worth noting that if the UAV enters the CFC during its flight, the discrete points of $|p_1 p_2|$ are not calculated. If the amount of collected data meets the requirements, the UAV $|p_3 p_4|$ and $|p_2 p_3|$ will no longer be calculated. However, the discrete points of the line segment from the position of the UAV to the destination will be calculated directly.

The initial path obtained by the Algorithm 2 does not consider the treatment of obstacles. Therefore, we continue to design a method based on expansion for the treatment of obstacles.

As shown in Figure 4, we first detect the initial path points generated based on the CFC. If these points are within the obstacle area, we move the direction of the vertical line to where these path points are located until a safe distance from the obstacle is found. This is accomplished through the following four steps: (1) Calculate the equation of the line for the neighboring points $\mathbf{p}_{initCFC}(i-1)$ and $\mathbf{p}_{initCFC}(i)$ on $\mathbf{p}_{initCFC}$; (2) calculate the distance $d_v$ from the center of the obstacle to $l_{\mathbf{p}_{initCFC}(i-1)\mathbf{p}_{initCFC}(i)}$; (3) determine if the distance $d_v$ is shorter than $r_{safe}$. Calculate the vertical line $l_{\mathbf{p}_{initCFC}(i)p_m}$ going through the point $\mathbf{p}_{initCFC}(i)$; (4) search for a point $p_v$ on the vertical line $l_{\mathbf{p}_{initCFC}(i)p_v}$ which satisfies that the distance from the obstacle center $p_m$ to $\mathbf{p}_{initCFC}(i-1)$ is greater than $r_{safe}$. Use that point as the new $\mathbf{p}_{initCFC}(i)$. The specific implementation steps are shown in Algorithm 3.

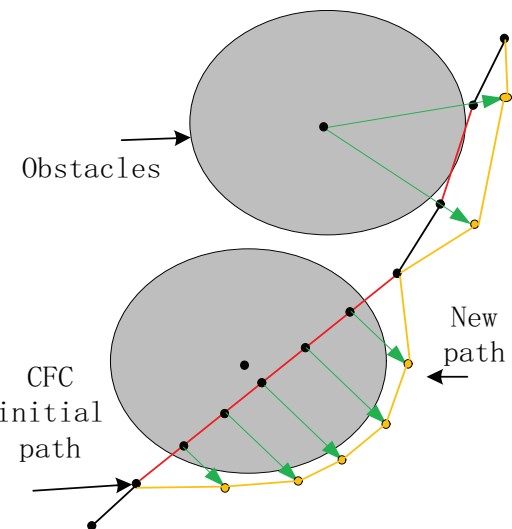

**Figure 4.** CFC initial path correction for obstacle avoidance.

---

**Algorithm 3:** For a given initial CFC path $\mathbf{p}_{initCFC}$, the position of obstacles $p_m$, the safe distance $r_{safe}$, and step factor $\lambda$ to find the initial global obstacle avoidance discrete path.

---

1: **function** *Initial_OA_path*$(\mathbf{p}_{initCFC}, p_m, r_{safe}, \lambda)$
2: **Loop for** $i = 2, 3, \cdots$
3:    Calculate the line going through points $\mathbf{p}_{initCFC}(i-1)$ and $\mathbf{p}_{initCFC}(i)$
4:    $a = [\mathbf{p}_{initCFC}(i) - \mathbf{p}_{initCFC}(i-1)]$
5:    $\Lambda = \begin{bmatrix} 0 & 1 \\ -1 & 0 \end{bmatrix}$
6:    $a^T \Lambda p = a^T \Lambda \mathbf{p}_{initCFC}(i)$
7:    **Loop for** $m = 1, 2, \cdots$
8:      **if** $\frac{|a^T \Lambda p_m - a^T \Lambda \mathbf{p}_{initCFC}(i)|}{\|a\|} \leqslant r_{safe}$ and $\|\mathbf{p}_{initCFC}(i) - p_m\| \leqslant r_m$
9:        Calculate the vertical line
10:        $a^T p = a^T \mathbf{p}_{initCFC}(i)$
11:        Take a point on the vertical line $p_v$
12:        Let $a_v = [p_v - \mathbf{p}_{initCFC}(i-1)]$
13:        **while** $\frac{|a_v^T \Lambda p_m - a_v^T \Lambda p_v|}{\|a_v\|} < r_{safe}$
14:          $p_v = p_v - \lambda \cdot \frac{p_v - \mathbf{p}_{initCFC}(i)}{\|p_v - \mathbf{p}_{initCFC}(i)\|}$
15:          and $a_v = [p_v - \mathbf{p}_{initCFC}(i-1)]$
16:        **endwhile**
17:       $\mathbf{p}_{initCFC}(i) = p_v$
18:      **endif**
19:    **End Loop**
20: **End Loop**
21: $\mathbf{p}_{initOA} = \mathbf{p}_{initCFC}$
22: return $\mathbf{p}_{initOA}$
23: **end**

---

### 3.4. Replanning

Based on the previously presented conclusions regarding the initialization path, we propose an SCA-based solution method for the convex optimization problem P1. The method approximates the optimal solution of the original problem P0 one by one through good initialization values and an iterative method based on Taylor expansions. The specific steps are shown in Algorithm 4. First, we initialize the state of the UAV and mission

settings. Subsequently, we generate the communication corridor path and safe flight path for obstacle avoidance as the initial values to solve the optimization problem P1 according to the mission requirements. Good initial values enable the iteration-based SCA algorithm to converge quickly and obtain a local solution. However, in the actual mission environment, the disturbance sources and obstacles may change dynamically, and the path determined at one point in time is not applicable to the whole flight process. For this reason, we propose a planning algorithm based on replanning in which the UAV plans a new trajectory at a certain frequency. The specific process is shown in Algorithm 5.

---

**Algorithm 4:** SCA-based trajectory planning method for (P1)

---

1: **Initialization:**
2:    Set the number of iterations $r = 0$.
3:    Set the initial state of the UAV, position, speed, acceleration, etc.
4:    Set the minimum required amounts of the communication bandwidth $B$ and data $Q_{th}$. Calculate the maximum communication distance $d_k^{\max}$ and maximum communication rate $r_{th}$ in accordance with the transmission system.
5:    Generate a CFC that satisfies the data collection conditions with Algorithm 1.
6:    Generate the initial CFC-based path with Algorithm 2.
7:    Generate an obstacle avoidance path based on the initial CFC path for a safe flight with Algorithm 3.
8:    Initialize $p^r[n]$, $\kappa_k^r[n]$, $\varepsilon_k^r[n]$, $\lambda_{i,m}^r[n]$, etc. with the path $\mathbf{p}_{initOA}$ generated by Algorithm 3, for $n = 1, \cdots, N$.
9: **Repeat**
10:    Solve the problem (P1) by CVX and acquire the optimal solutionsn $p^*[n]$, $\kappa_k^*[n]$, $\varepsilon_k^*[n]$ and $\lambda_{i,m}^*$
11:    Update the optimization variables and slack variables at the $r$-th iteration following:
12:      $w^{r+1}[n] = w^*[n]$;
13:      $\kappa_k^{r+1}[n] = \kappa_k^*[n]$;
14:      $\varepsilon_k^{r+1}[n] = \varepsilon_k^*[n]$;
15:      $\lambda_{i,m}^{r+1}[n] = \lambda_{i,m}^*[n]$;
16:    Update $r = r + 1$
17: **Until** some termination conditions are met

---

**Algorithm 5:** Receding horizon planning.

---

1: **Initialization:** Set the minimum communication bandwidth $B$ and data amount $Q_{th}$. Calculate the minimum communication rate $r_{th}$ and maximum communication distance $d_k^{\max}$ in accordance with the transmission system, maximum UAV speed $v_{max}$, maximum acceleration $a_{max}$, maximum flight height $H_{max}$, start position $p_0$, and end position. And let init state $s_{UAV} = [p_0, 0, 0]$.
2: **While:** $\|p_{uav} - p_N\| \leqslant \zeta$ **do**
3:    Update the UAV states
4:    Solving for global paths with Algorithm 5
5:    Select a section of path points for B-spline smoothing
6: **End**

---

To obtain a more suitable trajectory for the flight control module, we used B-splines to smooth the path in Algorithm 5. $N_Q$ path points were chosen as control points for the homogeneous B-spline curve according to the set replanning frequency, which is uniquely identified via its $p_b$, $N_Q$ control points $\{Q_1, Q_2, \ldots, Q_{N_Q}\}$ and a knot vector $\{t_1, t_2, \ldots, t_{M_Q}\}$, where $M_Q = N_Q + p_b$. To simplify and improve the efficiency of the trajectory assessment, the B-spline employed in our approach is homogeneous, which implies that each knot

has the same time interval $\delta$ between it and its predecessor. We normalize $t$ as follows: $s(t) = (t - t_m)/\delta$. The matrix representation can be used to assess the location [29]:

$$\mathbf{p}[s(t)] = \mathbf{s}(t)^T \mathbf{M}_{p_b+1} \mathbf{q}_m, \tag{64}$$

$$\mathbf{s}(t) = \begin{bmatrix} 1 & s(t) & s^2(t) & \cdots & s^{p_b}(t) \end{bmatrix}^T, \tag{65}$$

$$\mathbf{q}_m = \begin{bmatrix} Q_{m-p_b} & Q_{m-p_b+1} & Q_{m-p_b+2} & \cdots & Q_m \end{bmatrix}^T. \tag{66}$$

where $\mathbf{M}_{p_b+1}$ is a constant matrix determined by $p_b$. In our implementation, $p_b$ is set as 3, and the third-order $\mathbf{M}_{p_b+1}$ matrix has the following form:

$$\mathbf{M}_4 = \frac{1}{3!} \begin{bmatrix} 1 & 4 & 0 & 0 \\ -3 & 0 & 3 & 0 \\ 3 & -6 & 3 & 0 \\ 1 & 3 & -3 & 1 \end{bmatrix}. \tag{67}$$

*3.5. Convergence and Complexity Analysis*

Motion planning of a UAV in a data collection mission with dynamic jamming consists of five main parts: CFC construction, generation of the initial CFC path, generation of a safe path, SCA-based path optimization, and B-spline-based trajectory optimization. The convergence of the algorithm depends mainly on solving the P1 problem, which uses a method based on SCA. Section 2.1.2 in [30] presents an analysis of the convergence of SCA. We set $\bar{x}$ as the limit point of the iterative result produced via Algorithm 4, which met and slater condition holds at the point $\bar{x}$. Subsequently, $\bar{x}$ becomes a KKT point of P1. First of all, problem P1 indicates that there exist $5MN$ obstacle avoidance constraints of the UAV, $3N$ speed constraints, $3N$ initial position constraints, and $3N$ end position constraints, $6N$ equation of state constraints, and a $4NK$ quality of communication link or the amount of collected data. Hence, the computational complexity is approximately $O(R(5MN + 12N + 4NK))$, where $R$ represents the number of iterations, $M$ is the obstacle number, $N$ is the number of time slots, and $K$ is the SN number. In the construction process of the CFC, the goal is to acquire an effective communication area by applying geometric approaches. The complexity is only associated with the number of sensors and jammers. Thus, its complexity is $O(KJ)$. In the generation of the CFC initial path and safe path, the calculation volume is also related to the number of obstacles and number of sensors. Thus, the assistance is also $O(2KM)$. Finally, the B-spline trajectory only computes a local number of control points. Because we can ignore the number of operations, the total complexity is $O(R(5MN + 12N + 4NK) + 2KM + KJ + N)$. It is worth mentioning that motion planning in this paper is divided into global path planning and local trajectory optimization. The replanning time of global path planning is 1Hz; that is, it is updated once every second. In the hardware environment mentioned in reference [12], when the main frequency is 3 GHz, for a sensor, when the number of obstacles is 5, the number of interference sources is 2, the number of discrete points is 100, and the calculation time of the algorithm is about 3 ms. Therefore, it can fully support 1 s update frequency of path planning. At the same time, the local trajectory optimization directly selects the path points and adopts the B-spline method for fitting, which takes almost no time and is consistent with the literature [27]. The planning time is about 0.8ms, which can support the update frequency of 1000 Hz.

## 4. Simulations and Discussion

In this section, we present the results of numerical simulations to compare the algorithm in this paper with the SCA algorithm forinitialization proposed in [6] and the traditional DWA algorithm for dynamic environments. We mainly consider two dynamic sources of interference and several obstacles in the environment. To facilitate the simulation, we simulate a data collection task for an SN. Table A1 lists some crucial parameters

and their values. We made some changes to ensure that the SCA algorithm for AStar initialization from [6] is applicable to the scenario presented. In this paper, we divide the path into two segments: the paths between the origin and the SN and between the SN and the end point. The two paths are initialized separately with the A star algorithm. We further interpolate the A star initial path points in order to obtain initial conditions that can meet the required data volume. For the DWA algorithm, we design different cost functions depending on the stage of the task, as shown in Appendix A.

Figure 5 presents the motion plan for 200 kbit data collection in an environment without jamming and environments with static and dynamic jamming. In the jamming-free environment, the SN has the largest communication range, which covers the entire mission path of the UAV. In the static environment, the communication range of the SN is smaller and "pear-shaped"; the UAV can only collect data within this area. The communication range of the SN varies according to the motion of a jammer in a dynamic environment.

The red dashed line in Figure 6 represents the global path that is initialized via the CFC, the blue dashed line denotes the global path that is initialized via the AStar algorithm, and the green dashed line denotes the local path that is initialized through the DWA. The light green region is the SN's effective communication range; in the absence of interference, it should be a circle in the 2D top view. The effective communication range of the SN becomes "pear-shaped" when suppressed by two sources of interference. The grey circle is the obstacle, and the 3D shape is a cylinder. Evidently, both the red dashed and blue dashed lines are at safe distances from the obstacle and pass through the SN. This is exactly what we want: a safe initial path to collect data.

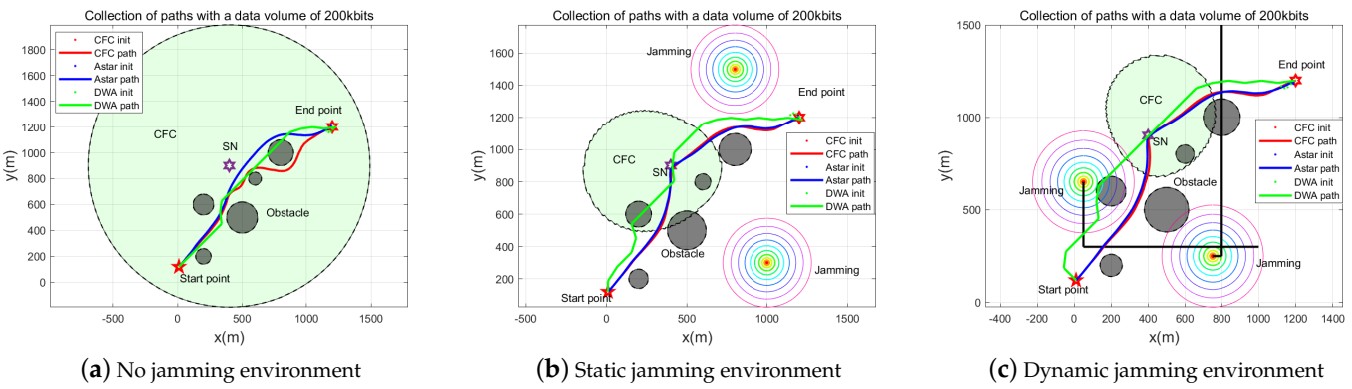

(**a**) No jamming environment     (**b**) Static jamming environment     (**c**) Dynamic jamming environment

**Figure 5.** 200 kbit data collection with three algorithms for motion planning in environments without jamming and with static and dynamic jamming.

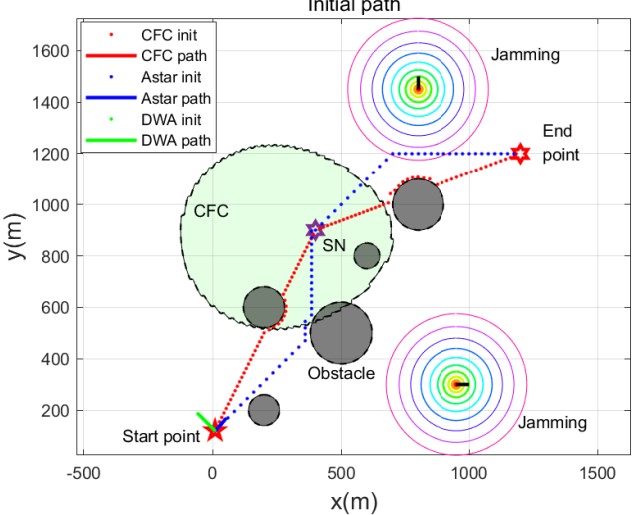

**Figure 6.** Initial path.

Figure 7 illustrates the residual amount of data collected by the UAV during the mission. In the jamming-free environment, the CFC and AStar algorithms complete the mission in approximately 25 s, while the DWA algorithm needs 30 s. In the static jamming environment, the CFC and AStar algorithms complete the entire mission in approximately 40 s, whereas the DWA algorithm needs 45 s. In the jamming interference environment, the CFC and AStar algorithms require similar times to complete the task, while the DWA algorithm needs more time.

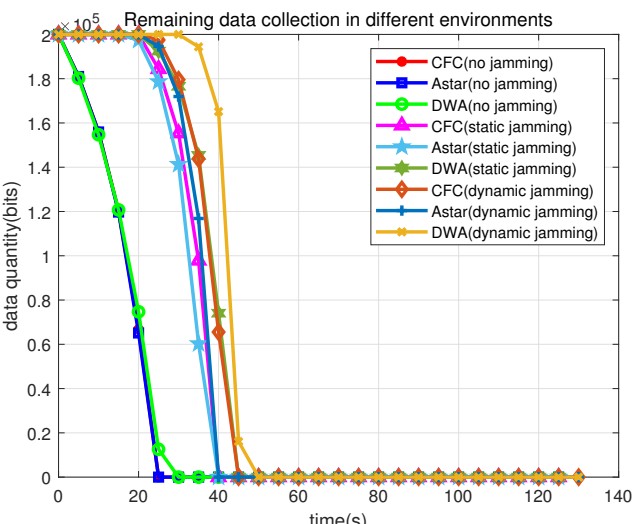

**Figure 7.** The residual amount of data collected for three algorithms in different environments.

Figure 8 shows the normalized energy consumption of the UAV. We used the integral of the UAV acceleration to present the UAV energy consumption. According to the figure, the DWA algorithm consumes much more energy than the CFC and AStar algorithms for the three environments. The AStar algorithm consumes slightly less energy than the CFC algorithm in the jamming-free environment, whereas the CFC consumes slightly less energy than the AStar algorithm in the jamming environment. It is also worth noting that the complexity of the AStar algorithm is $O(N^2 \log N)$, while the complexity of the CFC algorithm is only $O(KJ + 2KM + N)$. Therefore, collecting data with the CFC algorithm is much less expensive than with the AStar algorithm.

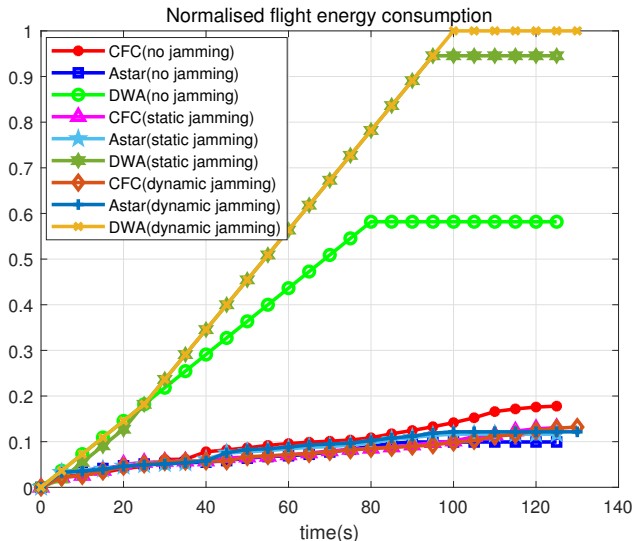

**Figure 8.** Variation in flight energy consumption for three algorithms in different environments.

The UAV trajectories for the different required data volumes are shown in Figure 9. Accordingly, the UAV trajectories planned for a required data volume of 0 bits ignore the communication corridors and interference. With growing amount of required data, the UAV trajectory stays in the communication corridor for a longer period of time.

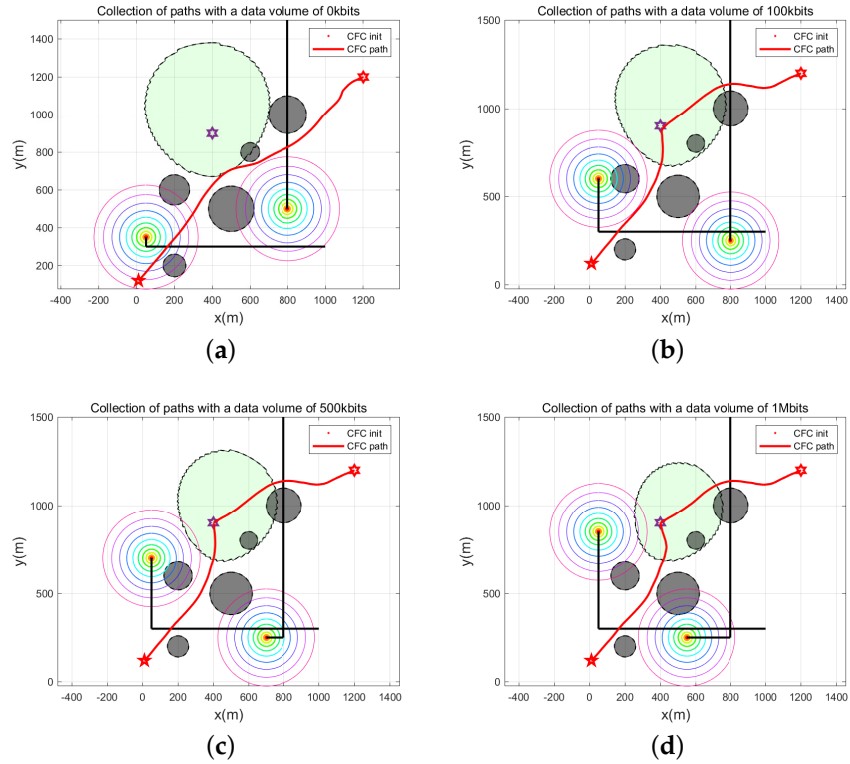

**Figure 9.** UAV paths for different data collection volumes: (**a**) 0 bits; (**b**) 100 kbits; (**c**) 500 kbits; (**d**) 1 Mbits.

Figure 10 shows the variation in the amount of remaining data for different collection requirements. The larger the amount of required data, the longer the task takes to be completed, which is in line with the general rule. Figure 11 shows the variation in the in-flight energy consumption for different collection requirements. The lowest energy is consumed for zero collected data, and the highest energy is consumed for the collection of 1 Mbit.

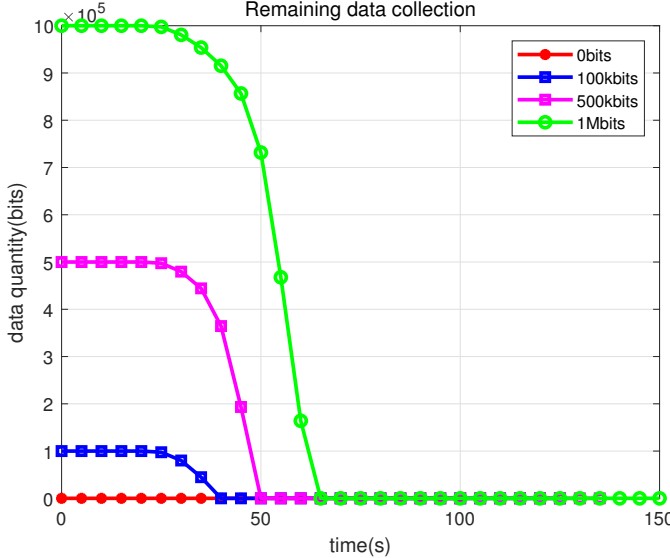

**Figure 10.** Variation in the amount of data remaining for different data collection requirements.

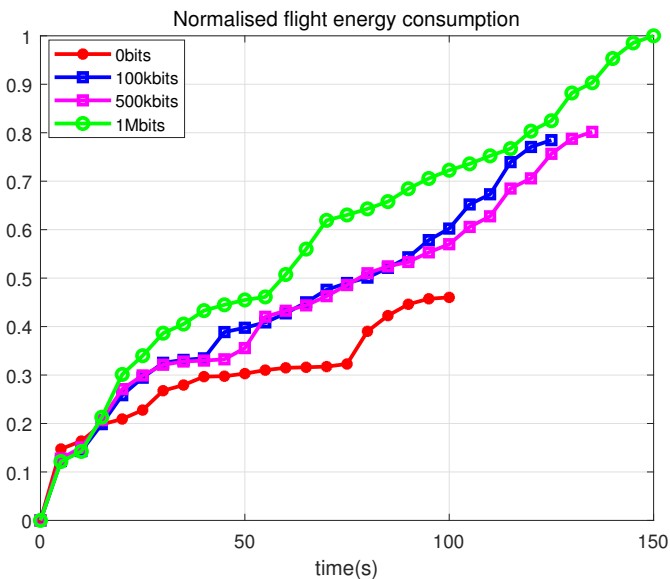

**Figure 11.** Normalized flight energy consumption.

Figure 12 displays the instantaneous communication rate of the UAV for different amounts of required data. The sky-blue dashed line indicates the communication threshold. Evidently, the greater the data demand exceeds the communication threshold area; for example, the red line in the figure is the instantaneous communication rate when the data collection volume is 0, which is basically less than the communication threshold. It does not need to consider the data collection task only needs to fly with the lowest energy consumption to focus. When the data collection volume is 1 Mbit (as shown by the green curve in the diagram), it will stay above the communication threshold as much as possible until the task has been completed.

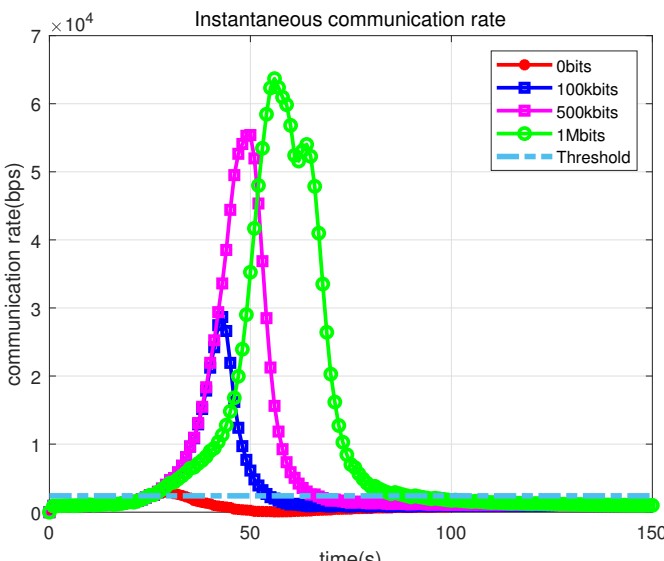

**Figure 12.** Instantaneous communication rate.

Figures 13 and 14 show the speed and flight altitude of the UAV during data collection for different required data volume, respectively. The UAV accelerates, decelerates, accelerates, and decelerates again, which is very much in line with the trapezoidal criterion we used during initialization. Therefore, the UAV should fly at full speed before entering the CFC, then decelerate appropriately to remain in the area long enough to collect the required amount of data in the CFC, and then fly at full speed to the end point. The flight

altitude also increases and then decreases, which is very much in line with the dynamics of a low-energy flight.

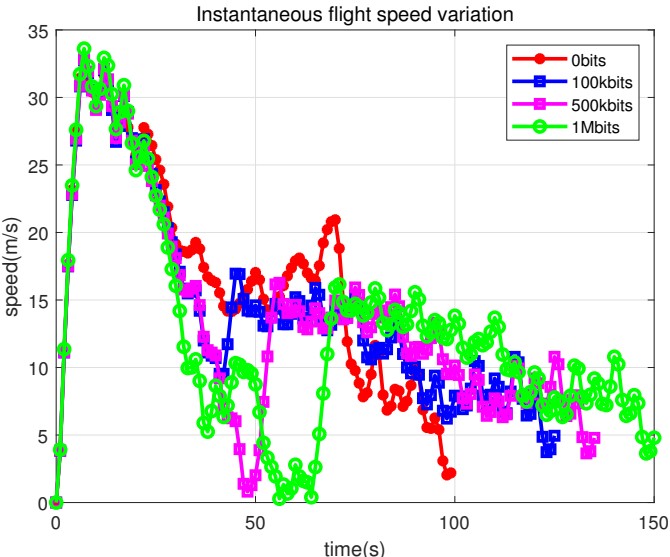

**Figure 13.** Instantaneous flight speed variation.

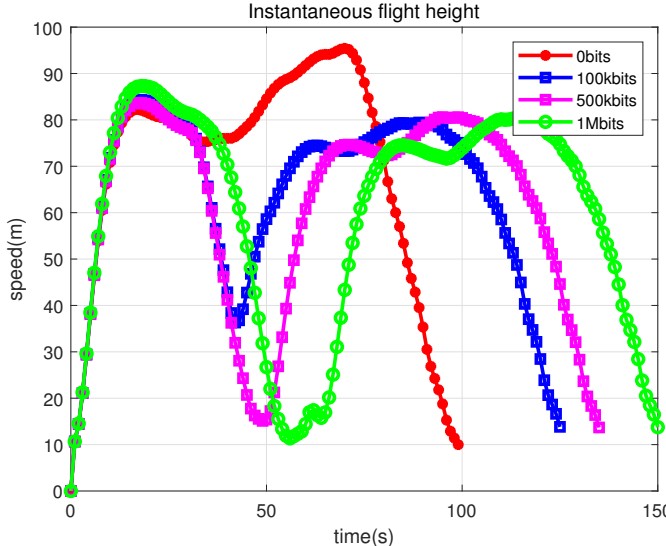

**Figure 14.** Instantaneous flight height.

## 5. Conclusions

In this study, the motion programming of UAVs collecting data in a dynamic jamming environment was examined. Our main optimization objective was to find a safe path that consumes a minimum amount of energy and results in the required amount of collected data. As the constructed optimization problem is non-convex, we introduced auxiliary and relaxation variables to relax the original problem and used the SCA algorithm to solve the new relaxation problem. To obtain a well-localized optimal solution with the SCA algorithm, we propose a fast initialization path method that is based on a CFC for processing. We constructed the CFC and solved for the initial path of the UAV with the UAV's perceived interference intensity and the transmission capability of the SN. Finally, we compared the simulation results with those of the AStar and DWA algorithms. The results demonstrate that the algorithm presented in the current paper is feasible and performs reliably.

**Author Contributions:** Methodology, D.G.; Software, C.X.; Formal analysis, B.Z.; Investigation, W.M.; Writing—original draft, B.W.; Funding acquisition, H.J. All authors have read and agreed to the published version of the manuscript.

**Funding:** This research received no external funding.

**Conflicts of Interest:** The authors declare no conflicts of interest.

## Appendix A

**Table A1.** Key simulation parameters.

| The Notation | Physical Meaning | Value |
|:---:|:---:|:---:|
| $P_0$ | SN transmission power | 500 mW |
| $P_j$ | Jammer transmission power | 5000 mW |
| $B$ | Total channel bandwidth | 10 kHz |
| $v_{max}$ | Maximum UAV speed | 25 m/s |
| $a_{max}$ | Maximum UAV acceleration | 5 m/s |
| $H_{max}$ | Maximum UAV height | 200 m |
| $H_{min}$ | Minimum UAV height | 10 m |
| $\sigma^2$ | Noise power spectral density | $-169$ dBm/Hz |
| $\alpha$ | Path loss exponent | 2 |
| $r_{th}$ | Minimum collection rate | 2.4 kbps |
| $\delta$ | Minimum update period | 1 s |
| $f_c$ | Communication carrier frequency | 2 GHz |

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
