# Peer review of "Motion Planning in UAV-Aided Data Collection with Dynamic Jamming"

_electronics, doi:10.3390/electronics12081841_

Round 1

Reviewer 1 Report

An optimization-based approach is adopted in this work to solve the calculation of the minimum-energy path for an UAV, under the complex constraints of data-collection and obstacle avoidance. The authors propose a convex relaxation to find the path, which is further refined to avoid the obstacles and ultimately smoothing the trajectory.

The paper is scientifically sound and it is correctly presented. The results support the theoretical work and show improvement over other methods. There are a few minor issues to enhance the quality of the paper, which -I suggest next:

1) Though the theoretical computational complexity of the algorithm is given in the paper, what is a typical execution time for solving the calculations of the trajectory? Can this be done in real time?

2) While Fig. 7 suggests that data collection finishes in around 50 s, Figs. 8 and 10-14 use a time span of around 150 s for the time flight. Please, explain this more clearly.

Author Response

The authors would like to thank the editor and all of the reviewers for their helpful and insightful comments. We have improved the quality of the manuscript by carefully taking all the comments into account. The modifications in this revised manuscript as well as the response to the editors’ and reviewers’ comments are described in the pdf file.

Reviewer 2 Report

see attachment

Author Response

The authors would like to thank the editor and all of the reviewers for their helpful and insightful comments. We have improved the quality of the manuscript by carefully taking all the comments into account. The modifications in this revised manuscript as well as the response to the editors’ and reviewers’ comments are described in pdf file.
